# Analysis of Predictive Coding Models for Phonemic Representation Learning in Small Datasets

**María Andrea Cruz Blandón** [1]   **Okko Räsänen** [1] [2]

## Abstract

Neural network models using predictive coding are interesting from the viewpoint of computational modelling of human language acquisition, where the objective is to understand how linguistic units could be learned from speech without any labels. Even though several promising predictive coding -based learning algorithms have been proposed in the literature, it is currently unclear how well they generalise to different languages and training dataset sizes. In addition, despite that such models have shown to be effective phonemic feature learners, it is unclear whether minimisation of the predictive loss functions of these models also leads to optimal phoneme-like representations. The present study investigates the behaviour of two predictive coding models, Autoregressive Predictive Coding and Contrastive Predictive Coding, in a phoneme discrimination task (ABX task) for two languages with different dataset sizes. Our experiments show a strong correlation between the autoregressive loss and the phoneme discrimination scores with the two datasets. However, to our surprise, the CPC model shows rapid convergence already after one pass over the training data, and, on average, its representations outperform those of APC on both languages.

## 1. Introduction

According to a number of influential neurocognitive hypotheses, the human brain uses predictive mechanisms for perception of and learning from sensory data (Friston, 2005; 2010; Cope et al., 2017). Similar ideas have been adapted

[1]Unit of Computing Sciences, Tampere University, Finland [2]Dept. Signal Processing and Acoustics, Aalto University, Finland. Correspondence to: María Andrea Cruz Blandón <maria.cruzblandon@tuni.fi>, Okko Räsänen <okko.rasanen@tuni.fi>.

*Published at the workshop on Self-supervision in Audio and Speech at the $37^{th}$ International Conference on Machine Learning*, Vienna, Austria. Copyright 2020 by the author(s).

to unsupervised neural network models, one of them the so-called Predictive Coding (PC) framework (see (Spratling, 2017) for a review of PC algorithms). Previously, PC has been used in image processing (Hénaff et al., 2019) and speech processing (Oord et al., 2018; Chung et al., 2019; Chung & Glass, 2019; Lian et al., 2019; Schneider et al., 2019).

The PC-based models are of special interest for low-resource speech technology, where access to labelled data is limited, but also for research on early language acquisition, where neurocognitively motivated approaches are of particular interest. In the latter, good models of human language learning should learn linguistic information from speech without any a priori linguistic specification. In both low-resource processing and modelling of human learning, the models should generalise across languages. Low-resource systems should also work with small datasets, whereas high-quality datasets used to study language learning are also often limited in size. One of the resulting challenges is the application of the same models across the different corpora, where a good system would require little if any hyperparameter optimisation across the different use cases. Since hyperparameter optimisation is time-consuming and often not feasible, the use of conventional hyperparameters is common.

In this paper, we examine the performance of PC models applied to learn of phonemic representations from speech in the context of two new languages, French and Mandarin, whose corpora are also smaller compared to the original studies. The work contributes to the understanding of these models, and provides support for model selection when applying these models to real low-resource scenarios. We focus on three questions: a) is there a consistent relationship between the model loss functions and phoneme selectivity of the learned representations across different datasets, b) how much is this relationship affected by the dataset type and size, and c) how does learning in these models compare a function of the amount of training data available?

## 2. Predictive Coding Models

In this section, we will explain the two selected PC models, APC (Chung et al., 2019) and CPC (Oord et al., 2018). The fundamental difference between the two is the optimisation

problem that each model tries to solve. More specifically, APC uses an autoregressive loss trying to predict future input features accurately while CPC uses a contrastive loss that focuses on distinguishing real future latent representations from false future. The authors of APC argue that there is evidence that a low contrastive loss implies the existence of a classifier with a low unimodal loss (Chung et al., 2019). In contrast, in CPC, the authors claim that unimodal losses are not convenient when we want the model to excel in capturing the relationships between the data and its context in high dimensional data such as time-frequency structure of speech (Oord et al., 2018).

### 2.1. Contrastive Predictive Coding

The underlying motivation for CPC is to extract information from the temporal context that serves to describe the data more effectively. To achieve this, CPC's authors propose a model that aims to maximise the mutual information between the data and its future context.

The architecture comprises two blocks. In the first block, a non-linear encoder processes the input features (raw audio waveform in the original paper). The outputs of this block are called the latent representations, $\mathbf{z}_t$. This block is followed by an autoregressive block that produces so-called context latent representations $\mathbf{c}_t$ using the history of previous latent representations $\mathbf{z}_{\leq t}$. Using $\mathbf{c}_t$, the model predicts latent representations $k$ time steps ahead using $\mathbf{z}'_{t+k} = \mathbf{W}_k\mathbf{c}_t$, which correspond to the predictive coding part.

To maximise the mutual information between input features and context representations, the authors introduce InfoNCE loss. This loss is based on Noise-Contrastive Estimation (NCE) (Gutmann & Hyvärinen, 2010). Assuming there is a noise distribution close to the data distribution, the model can learn by comparison. The model reaches this aim by discriminating the samples taken from the data distribution and the ones taken from the noise distribution, which are called negative samples. In CPC, the negative samples are randomly taken from the data distribution as in (Bengio & Senécal, 2008). The InfoNCE loss corresponds to the categorical cross-entropy loss (see Eq. (1)), where a density ratio gives the score of the sample classification. The model does not require to learn the probabilistic data distribution directly, instead uses a log-bilinear model for the density ratio, $f_k(\mathbf{x}_{t+k}, \mathbf{c}_t) = exp(\mathbf{z}_{t+k}^T\mathbf{W}_k\mathbf{c}_t)$.

$$L_{\text{InfoNCE}} = -log\frac{f_k(\mathbf{x}_{t+k}, \mathbf{c}_t)}{\sum_{\mathbf{x}_j \in \mathbf{X}} f_k(\mathbf{x}_j, \mathbf{c}_t)} \tag{1}$$

### 2.2. Autoregressive Predictive Coding

Based on the hypothesis that a low contrastive loss implies the existence of a linear classifier with a low unimodal loss

(Chung et al., 2019), authors of APC propose an autoregressive model for the PC. APC is similar to autoencoder architectures in which the target features are the same as the input features, except that in APC, the target features are the input features occurring in future time steps.

APC architecture consists of a 'PreNet' block that maps the input features (80-dim log Mel spectrograms in the original paper) to a new vector space, an autoregressive model, and a 'PostNet' block implementing the PC part. The 'PostNet' block predicts the future $k$ features $\mathbf{x}_{t+k}$, using the latent representation ($\mathbf{z}_t$) output by the autoregressive model. As a result, the model learns the probability distribution of future features. APC uses the Mean Absolute Error (MAE) as the loss function to optimise the training (see equation 2), where $y_{t+k}$ is the prediction for the signal $x_{t+k}$. Therefore, the latent representations should then encode information that helps the model to reconstruct the input features $k$ steps in the future.

$$L_{\text{MAE}} = \frac{\sum_{t=1}^{N-k} |\mathbf{x}_{t+k} - \mathbf{y}_{t+k}|}{N - k} \tag{2}$$

## 3. Experimental Setup

In this section, we describe the corpora, model architectures, and the experimental setup we used to analyse the relationship between APC and CPC validation losses and their performance in a phoneme discrimination task.

### 3.1. Datasets and phoneme discrimination tasks

We tested APC and CPC models on a subset of the track 1 of the Zero Resource Speech Challenge 2020 datasets (Dunbar et al., 2017) that focuses on learning of phoneme-sensitive features in an unsupervised manner. The subset contains 24 h of French and 2.5 h of Mandarin conversational speech for model training, and $47,096$ and $21,247$ one second utterances for testing in the two languages, respectively. The training datasets are composed of a few speakers with more speech (approx. 20 min for Mandarin and 2 h for French), and several speakers with short recordings (about 10 min each). We tried to maximise speaker diversity (unique speakers) in the training while maintaining train/validation split ratio of $80\%/20\%$ as closely as possible.

In the context of the challenge, the task consists of learning speech representations that are convenient for phoneme discrimination, for which the challenge incorporates a minimal pair ABX-task (Schatz et al., 2013; 2014). The task measures the phonemic discriminability of the learned representations (Versteegh et al., 2015; Dunbar et al., 2017). In our experiments, the evaluation tool provided by the challenge was used to calculate the ABX scores. ABX scores are reported separately for *within-speaker* (minimal pair tokens always from the same talker) and *across-speaker* conditions

(tokens from different speakers), where the latter better reflects speaker-independent phonemic categorisation.

### 3.2. Implementation of the PC models

As input features, 39 MFCC (13 static $+ \Delta + \Delta\Delta$) coefficients were extracted using a window length of 25 ms and a window shift of 10 ms. The data was split into 2 s samples. For each epoch, the order of the input data was randomised. All models were trained in a monolingual setup.

For APC, we followed the implementation published by (Chung et al., 2019). The network consists of three fully connected layers with 128 units with ReLU activations for 'PreNet' with 20% of dropout, three GRU layers with 512 units and residual connections (Wu et al., 2016) for the predictive part, and one convolutional layer with kernel size of one for the 'PostNet'. We used an initial learning rate of $10^{-4}$ unless otherwise specified. The prediction was carried out five frames (50-ms) ahead.

For CPC, we followed the implementation provided by (Schneider et al., 2019) for the contrastive loss calculation. We also followed the adaptation of the encoder proposed by (Chung et al., 2019) for using acoustic features as input features. The architecture consists of three fully connected layers with 512 units with ReLU activations for the encoder, and one GRU layer with 256 units for the autoregressive model, both blocks trained with a dropout of 20%. As in (Schneider et al., 2019), we used ten negative samples taken from the batch and predicted 12 steps frames, that is 120 ms ahead. We used an initial learning rate of $10^{-3}$.

We trained all models using a batch size of 32, and using Adam optimiser (Kingma & Ba, 2015). For all models, we used PCA to reduce the dimension of the latent vectors used for ABX task (maintaining 95% of the variance), as the original dimensionality was too high for the ABX-scoring tool to handle. The reported ABX-scores correspond to the extracted latent representations $\mathbf{z}_t$ for both models. Although context latent representations $\mathbf{c}_t$ were also analysed for the CPC model, we only report latent representations as there were no notable differences between $\mathbf{c}_t$ and $\mathbf{z}_t$.

### 3.3. Experiments

To assess the correlation between the validation loss and the ABX scores, the APC and CPC models were trained for 100 epochs and saving the models every ten epochs for ABX-scoring ('APC-1' and 'CPC-1'). Each model was trained three times with random initialisation to consider the influence of initial parameters. In the case of CPC, we ran an additional experiment ('CPC-2') to investigate the behaviour of the model during the first ten epochs in more detail, saving after each of the first 10 epochs and then every 10 epochs, and running the experiment twice.

*Table 1.* Percentage of the French dataset used for training. The number of hours that the percentage represents, and the number of samples for the training set (T.) and for the validation set (V.)

| PERCENTAGE | HOURS | T. SAMPLES | V. SAMPLES |
|---|---|---|---|
| 100 | 25.1 | $36,031$ | $9,182$ |
| 75 | 18.8 | $27,023$ | $6,886$ |
| 50 | 12.6 | $18,015$ | $4,591$ |
| 25 | 6.3 | $9,007$ | $2,295$ |

To calculate the correlation, Pearson's correlation coefficient ($r$) was adopted; however, in cases were the linear correlation was not evident in the scatter plot, we also calculated Spearman's rank correlation coefficient ($r_s$). Additionally, the significance of the correlation coefficients was validated performing a hypothesis test for $r$ and using the critical value (Zar, 1972) for $r_s$. In both cases with a significance level of $\alpha = 0.05$ (critical values equal to $r_s = 0.678$, and $t = 1.86$). The $t$ test statistic for $r$ was calculating with the formula $t = r\sqrt{n-2}/\sqrt{1-r^2}$, where $n$ is the number of points used for calculating $r$.

Regarding the relationship between the dataset size and the performance of the predictive model, we train four models varying the percentage of samples for the training data from 100% to 25% decreasing on 25% each time. For this analysis, the French dataset was employed, see table 1.

## 4. Results

Fig. 1(a) shows the validation loss and the ABX-scores of the APC model for the French and Mandarin datasets (APC-1). A striking correlation between the two values can be seen for the two languages; although the slope for Mandarin data is higher than for French data. There is also more variability in the French runs ($r = 0.817 \pm 0.076$ for ABX across-speaker; $r = 0.725 \pm 0.159$ for ABX within speaker) than in the Mandarin dataset ($r = 0.991 \pm 0.005$ for ABX across-speaker; $r = 0.978 \pm 0.009$ for ABX within-speaker). Since the French training started to overfit already after 20 epochs (with increasing validation loss), we re-ran these experiments for French dataset but using a lower learning rate ($lr = 10^{-5}$) (APC-2). As a result, the variability among the runs was reduced ($r = 0.997 \pm 0.001$ for ABX across-speaker and $r = 0.809 \pm 0.248$ for ABX within speaker. See Supplementary Material, Fig. S1 for the scatter plot).

In the first experiment for CPC (CPC-1), there was little relative variation in both the InfoNCE loss and the ABX-scores. A closer analysis revealed that the validation loss was decreasing with more epochs, whereas the ABX-scores were oscillating with small changes (standard deviation for the three runs: $SD = 0.217$ for ABX across-speaker for Mandarin; $SD = 0.318$ for ABX within-speaker for Man-

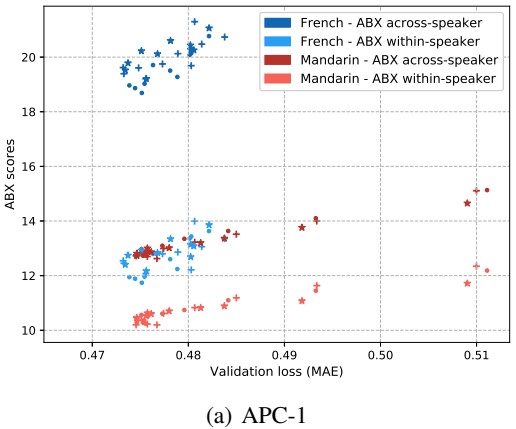

(a) APC-1

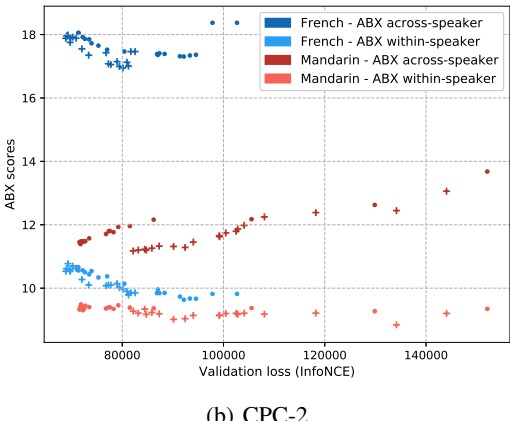

(b) CPC-2

*Figure 1.* Scatter plots of the APC-1 and CPC-2 models ABX performance as a function of validation loss, including a detailed picture of the first ten epochs for the CPC model. Symbol markers: (+) First run, (·) Second run, and (*) Third run.

darin; $SD = 0.145$ for ABX across-speaker for French; $SD = 0.174$ for ABX within-speaker for French). This behaviour suggested the model was converging to phoneme-like representations already in the first ten epochs. To evaluate this hypothesis, we ran a second experiment also evaluating all the models from the ten first epochs. To our surprise, the CPC model shows a rapid convergence after one pass over the training data (see ABX-scores for larger values of the validation loss). The oscillation pattern observed in the previous experiments persists with later epochs, and the change in overall ABX-score is nearly zero for almost all cases except for Mandarin ABX across-speaker condition, where a slight improvement is observed with more training. Notably, the CPC ABX performance after one epoch is already comparable to the APC best results.

Table 2 lists the correlation coefficients calculated for the averaged performance of the APC model (Mandarin APC-1 and French APC-2). Since the relationship between the InfoNCE loss and the ABX-scores is highly variable for the CPC model across the runs, we calculated the correlation coefficients for each run (CPC-1 (run id)). All APC correlation coefficients were found to be significant with significance criterion of $\alpha = 0.05$ ($t = 42.260$ for APC-2 FR ABX across-speaker; $t = 6.637$ for APC-2 FR ABX within-speaker; $t = 22.923$ for APC-2 MA ABX across-speaker; $t = 13.461$ for APC-2 MA ABX within-speaker). The CPC model, on the other hand, shows both positive and negative correlation for the same language (see, e.g., $r$ of ABX across-speaker score for CPC-1 (1,3) MA). This remarkable discrepancy highlights the variability between runs when the model has rapidly converged.

Table 3 shows the correlation coefficients obtained for the CPC model for the first ten epochs (CPC-2) for both languages. The relationship between the validation loss and the ABX across-speaker score shown in Fig. 1(b) was also reflected in the correlation coefficients obtained. Both $r$ and $r_s$ are significant and exhibit a strong positive correlation

*Table 2.* Correlation coefficients between the validation loss and the ABX-scores for the French (FR) and Mandarin (MA) datasets. Pearson's ($r$) and Spearman's Rank ($r_s$) correlation coefficients are reported for ABX-scores. (*) $\rho < 0.05$. Analysis of APC averaged performance and CPC runs.

| MODEL | ACROSS-SPEAKER | | WITHIN-SPEAKER | |
|---|---|---|---|---|
| | $r$ | $r_s$ | $r$ | $r_s$ |
| APC-2 FR | **0.998**$^*$ | 1.000$^*$ | **0.920**$^*$ | 0.879$^*$ |
| APC-1 MA | **0.992**$^*$ | 0.903$^*$ | **0.979**$^*$ | 0.867$^*$ |
| CPC-1 (1) FR | -0.202 | -0.115 | -0.703$^*$ | -0.770$^*$ |
| CPC-1 (2) FR | 0.920$^*$ | 0.867$^*$ | 0.836$^*$ | 0.588 |
| CPC-1 (3) FR | -0.511 | -0.661$^*$ | -0.228 | -0.055 |
| CPC-1 (1) MA | -0.705$^*$ | -0.552 | -0.525 | -0648$^*$ |
| CPC-1 (2) MA | 0.282 | 0.006 | -0.759$^*$ | -0.782$^*$ |
| CPC-1 (3) MA | 0.913$^*$ | 0.782$^*$ | 0.310 | 0.430 |

throughout the training ($r(8) = 0.972, \rho < 0.05$ for the first and $r(8) = 0.960, \rho < 0.05$ for the second run). The strong correlation for the ABX across-speaker score also shows a feature of the InfoNCE loss that is worth noting, although it was exhibited for some runs only. The selection of the negative samples could have an impact on the information that is favoured in the representations (Oord et al., 2018; Chung et al., 2019). The rationale behind this is that by using the same utterance to extract the negative samples, the information about speaker features will not be relevant for distinguishing true and negative samples, thus encouraging phonemic information. We run additional experiments to evaluate if the ratio change (relative proportion of change between consecutive epochs) for the validation loss was correlated to the ABX-scores, but our results did not provide statistical evidence of such correlation.

As for the dataset size comparison, Table 4 shows the ABX-scores obtained after training the APC model with different dataset size for the French language. Unlike earlier, the model was trained with a learning rate of $10^{-5}$, as this

*Table 3.* Correlation coefficients for the first ten epochs of the CPC model. $(*)$ $\rho \geq 0.05$.

| MODEL | ACROSS-SPEAKER | | WITHIN-SPEAKER | |
|---|---|---|---|---|
| | $r$ | $r_s$ | $r$ | $r_s$ |
| CPC-2 (1) FR | 0.218* | 0.219* | -0.869 | -0.851 |
| CPC-2 (2) FR | 0.795 | 0.255* | -0.323* | -0.608* |
| CPC-2 (1) MA | **0.948** | **0.988** | -0.406* | 0.285* |
| CPC-2 (2) MA | **0.957** | **0.964** | -0.587* | -0.479* |

*Table 4.* Performance of the APC model as a function of the dataset size.

| PERCENTAGE | ACROSS-SPEAKER | WITHIN-SPEAKER |
|---|---|---|
| 100 | 19.265 | 12.790 |
| 75 | 19.921 | 13.202 |
| 50 | 19.878 | 12.879 |
| 25 | 20.358 | 13.074 |
| $\bar{x} \pm SD$ | 19.856±0.449 | 12.986 ± 0.186 |

was found to improve training stability in the earlier experiments. Considering the strong correlation between MAE and the ABX-scores, each model was chosen based on the lowest validation loss. The differences in the ABX-scores are relatively negligible when taking into account that the models were trained for a maximum of 100 epochs (usually with the lowest validation loss value). This implies that the models could still improve their representations with more training. That being said, with only 25% of the total data, that is 6.3 h of the French dataset, the APC model already converged with the hyperparameters here defined. Contradictory to the idea that more training data improves the performance, this result shows that hyperparameter tuning would be more beneficial in this case than increasing the training data. For CPC, it was problematic because we could not use the validation loss as the selection criterium, and we could not conduct the experiments in time. However, see the supplementary material for an upper bound of the true performance assuming a rapid convergence.

As a final comparison, Table 5 lists the best ABX-scores obtained for the APC-1 and CPC-1 models, and the training epoch for which the best model was obtained. We also report CPC-2 model only after one epoch of training to demonstrate its fast learning. MFCC-based ABX-scores are also reported as a baseline. Both PC models improved the ABX-scores in comparison with the baseline, except for Mandarin ABX within-speaker score. The CPC model outperforms the APC model in both languages and ABX-scores.

## 5. Discussion and Conclusions

In this paper, we analysed the behaviour of PC models in the context of phoneme discrimination tasks with relatively

*Table 5.* Best ABX-scores obtained for the APC and CPC models among all the three runs of the first experiment and ABX-scores of the CPC model in the first epoch of the second experiment. In bold the lowest scores.

| MODEL | EPOCH | ACROSS-S | WITHIN-S |
|---|---|---|---|
| APC-1 FR | 10 | 18.698 | 11.740 |
| APC-1 MA | 100 | 12.624 | 10.197 |
| CPC-1 FR | 10 | 17.500 | **9.791** |
| CPC-1 MA | 20 | **11.837** | 9.185 |
| CPC-2 FR | 1 | **17.463** | 9.854 |
| CPC-2 MA | 1 | 13.058 | 9.202 |
| MFCC FR | - | 21.050 | 10.150 |
| MFCC MA | - | 14.584 | **9.140** |

small datasets. Our experiments confirmed that APC and CPC models are also suitable for relatively small corpora. In the original papers, the APC and CPC models were trained on 100- and 360-hour subsets from Librispeech (Panayotov et al., 2015), respectively. Our results show that these models also learn phoneme-discriminating representations from much smaller corpora down to mere 2.5 hours of speech.

A very high and consistent correlation ($r \approx 0.97$) between the MAE loss and ABX scores was found for the APC model across the two datasets. However, this correlation was affected by the sampling of epochs for the ABX evaluation, where a large proportion of the scores were obtained after the model had already saturated in performance. Despite this effect, which could easily be avoided by using early stopping, the APC behaves similarly for both datasets.

On the contrary, there was no significant correlation between validation loss and ABX scores for the CPC model. In fact, our results suggest that the CPC model was rapidly converging to effective phoneme-sensitive representations already during the first ten epochs. After this, the model continues learning representations that improve the predictive loss, but this is not reflected in better phonemic representations. The latter requires further experiments to understand the underpinning of this behaviour. Interestingly, the very good CPC performance already after one pass over the training data resembles the conditions of human language acquisition, where a child never has access to the same input twice.

Finally, APC results are especially important as they could be interpreted as evidence of adaptability to different dataset sizes and robustness to different languages; the validation loss can be employed for selecting the model when extracting phonemic features for different datasets. On the other hand, although the CPC model obtained the best ABX scores in early iterations, its validation loss is less directly linked with the phonemic nature of the learned representations in the case of small datasets.

## Acknowledgements

This study was funded by Academy of Finland grants no. 314602 and 320053.

## S1. Supplementary Material

### S1.1. Code and statistical data

Our implementation of the APC and CPC model and all the data points and statistical metrics could be found on https://github.com/SPEECHCOG/pc_models_analysis

### S1.2. Scatter plots

Figure S1 shows the APC-2 experiment for the French dataset. Figure S2 illustrates the CPC-1 experiment, three runs for each language with 100 epochs per run, and figure S3 is the detailed view of the ABX across-speaker scores over epochs for the three runs of the French dataset.

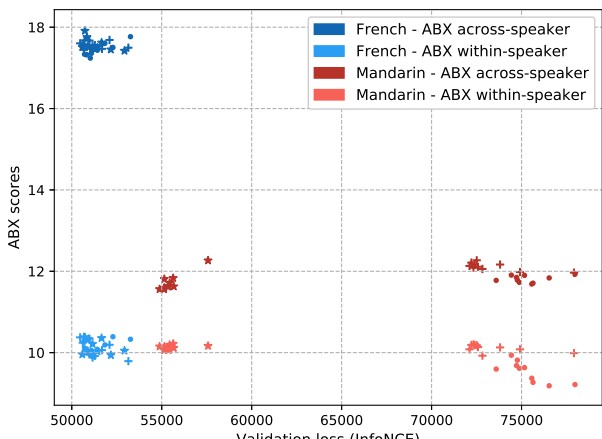

*Figure S2.* Scatter plot of the CPC model ABX performance as a function of the validation loss (CPC-1). Symbol markers: (+) First run, (·) Second run, and (*) Third run.

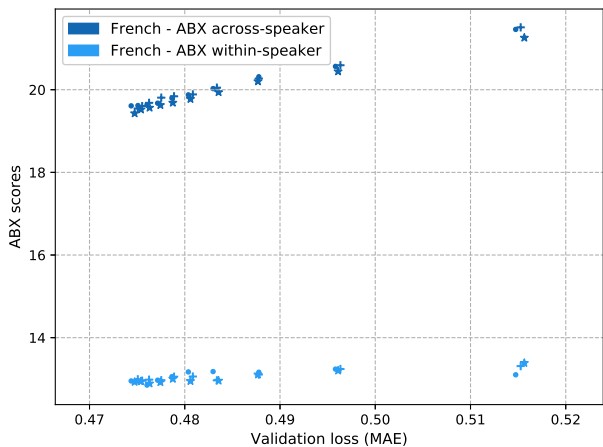

*Figure S1.* Scatter plot of the French APC model ABX performance as a function of the validation loss (APC-2). Model trained with $lr = 10^-5$. Symbol markers: (+) First run, (·) Second run, and (*) Third run.

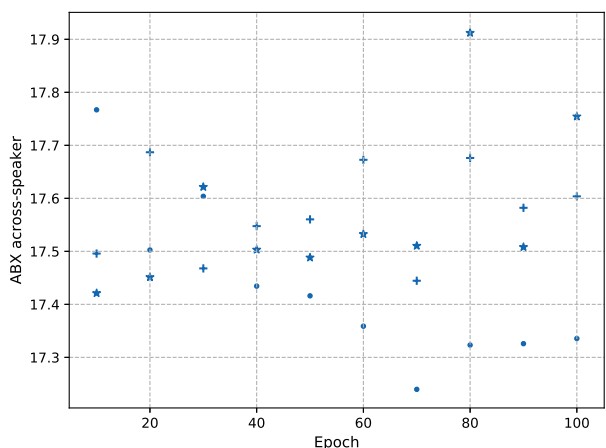

*Figure S3.* ABX across-speaker scores as a function of the epoch for the three runs of the French CPC model (CPC-1). Symbol markers: (+) First run, (·) Second run, and (*) Third run.

### S1.3. CPC dataset size experiment

In the case of the CPC model, there was not a significant correlation between the validation loss and the ABX-scores. As a consequence, it was less accurate to use the validation loss as the selection criterium of the model than for the APC model. To offer an upper bound of the real performance of the CPC model, we ran the dataset size experiment (see subsection 3.3) assuming a rapid convergence. For this experiment, we used the same architecture as explained in subsection 3.2. Table S1 shows the ABX-scores obtained after training the model for ten epochs with different dataset size for the French language. As in APC, by using roughly six hours of the French dataset (25%) the model obtained ABX-scores comparable to the ABX-scores obtained with the full dataset.

On the other hand, unlike the APC model, the ABX across-speaker score shows a slight improvement by increasing the dataset size. The infoNCE loss benefits from more data for the comparison of negative and true samples resulting in more speaker-independent phoneme representations. However, notice that the differences in the ABX within-speaker scores are relatively negligible. This behaviour is

*Table S1.* Performance of the CPC model as a function of the dataset size. Assuming a rapid convergence in 10 epochs.

| PERCENTAGE | ACROSS-SPEAKER | WITHIN-SPEAKER |
|---|---|---|
| 100 | 16.872 | 10.325 |
| 75 | 17.535 | 11.166 |
| 50 | 17.778 | 10.361 |
| 25 | 18.406 | 10.478 |
| $\bar{x} \pm SD$ | 17.648±0.634 | 10.583 ± 0.394 |

comparable to the results for the Mandarin CPC-2 models, where the ABX across-speaker score was improving over time, whereas the ABX within-speaker score was oscillating around the same value (see figure 1(b)). Further experiments are necessary to understand this behaviour.

### S1.4. APC with Mean Square Error loss

To explore the behaviour of the APC model with a different unimodal loss, we ran an extra experiment utilising the Mean Square Error (MSE) loss for training the model. Similar to previous experiments, we ran the model three times for 100 epochs and evaluated the performance on the ABX task every ten epochs for the Mandarin dataset.

Figure S4 shows the APC model ABX performance as a function of the MSE loss. The behaviour is comparable to APC with MAE loss. The Pearson's correlation coefficients are $r = 0.953 \pm 0.014, \rho < 0.05$ for ABX across-speaker score and $r = 0.908 \pm 0.005, \rho < 0.05$ for ABX within-speaker score. These results expose a high correlation between the ABX-scores and the MSE loss.

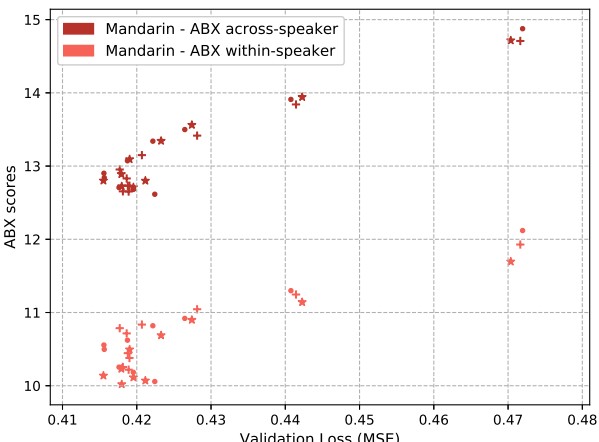

*Figure S4.* Scatter plot of the Mandarin APC model ABX performance as a function of the Mean Square Error loss. Model trained with $lr = 10^{-}4$. Symbol markers: (+) First run, (·) Second run, and (*) Third run.

In order to compare the correlation coefficients of the two

APC models (with MAE loss and with MSE loss), we performed a Z-test. We set the level of significance to $\alpha = 0.05$ indicating a critical value of $\pm 1.96$ and employed Fisher's transformation for the correlation coefficients of the averaged performance (APC (MSE): r=0.956 for ABX across-speaker and r=0.915 for ABX within-speaker; APC (MAE): r=0.992 for ABX across-speaker and r=0.979 for ABX within-speaker. All coefficients with $\rho < 0.05$). The observed Z values are $Z_{\text{obs}} = -1.672$ for ABX across-speaker and $Z_{\text{obs}} = -1.326$ for ABX within-speaker. We did not find sufficient evidence to conclude a significant difference between the correlation coefficients of the APC (MAE) model and the APC (MSE) model.

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
