# OpenReview forum: "Analysis of Predictive Coding Models for Phonemic Representation Learning in Small Datasets"
_ICML.cc/2020/Workshop/SAS — SAS 2020_

### Official Review · AnonReviewer3 · 2020-06-29
**Investigating APC and CPC learnt from smaller corpora**

**Rating:** 7
**Confidence:** 5

**Review:**

While self supervised learning is generally done from large amounts of untranscribed speech (many experiments pretend that Librispeech - 1000h - is untranscribed and learn models from it), this paper investigates two self supervised learning methodes (autoregressive predicting coding - APC and contrastive predicting coding - CPC)  learnt from smaller corpora (in French and Mandarin).
Experiments are done in the framework of the Zero resource Speech Challenge (ZRC), subword (phoneme) discovery task.
Also, the questions asked by the authors are the following: is there correlation betwwen CPC/APC learning loss and phoneme selectivity of learnt representations ? What is the impact of the amount of training data available .
The answer to first question shows that overall APC's learning loss is well correlated with phone discriminability (measured with ABX score) while surprisingly CPC is not. CPC seems to learn after few steps a good phoneme selectivity but after that, loss continues to decrease while phoneme selectivity does not improve anymore. This raises another question: what does the model learn after these few steps ? And why CPC's behavior is that different from APC ? But this question is not addressed in the paper (would be interesting to discuss these questions at least a little bit).
As far as the second question is concerned, I found less systematic experiments to show the impact of the amount of training data on self supervised learning. Only results for APC seem to be given on that aspect, not for CPC.


Overall this paper is well written and clear. I only have the following minor comments:
-3.1: 1st paragraph, did you change the train/dev split of ZRC ? (seems this is the case from the text)
=>then it's a pity since your ABX scores won't be comparable with "official" ones
-table 4: why don't you provide results for CPC as well ?

---

### Official Review · AnonReviewer2 · 2020-06-29

**Rating:** 7
**Confidence:** 5

**Review:**

The authors perform careful experiments with CPC [1] and APC [2] on top of MFCC features, and find that CPC converges much more quickly to representations that are useful for phoneme discrimination. This is a cool and surprising result---I would have thought that reconstructive AR modeling is always better than contrastive AR modeling, given the same input features, since the supervision signal is a rich feature vector instead of just a 1-bit "fake or real" label.
__________
Here is my main issue with this paper: what the authors call "CPC" is actually pretty different from its original presentation in [1], in a way that I think matters and I'll explain.

The idea behind CPC is this: we would like to do autoregressive modeling to learn a feature extractor. This is possible (e.g. WaveNet turned out to be a good feature extractor for TIMIT), but AR modeling is extremely expensive for high-dimensional signals like raw audio samples and pixels. What we could do is train the AR model in some lower-dimensional latent space, instead.

So in CPC, we encode the audio into a shorter sequence of feature vectors using a convolutional net, and then train the AR model on top of those shorter sequences. Here's the problem: there is nothing preventing the model from learning to encode the input signal as something really easy for the AR model, like all zeros. CPC gets around this problem by using a contrastive model: it must guess whether an encoding is real or fake (the actual next output of the encoder or a negative sample).

APC takes a different approach to solving the same problem. Instead of encoding the input signal using a trainable model, it uses good old FBANK features as the input to the AR model. The FBANK feature extractor is not trainable, so the encoder cannot learn to output something dumb. Now we can just do normal reconstructive AR modeling, as opposed to contrastive AR modeling. The tradeoff is that now we need handcrafted features and can't learn from the raw input signal, but we already know that FBANK features work well, and it outperforms the original CPC on a bunch of tasks.

So I think the fact that the raw input signal is used in CPC is a really crucial aspect of it---after all, the title of the original paper is "Representation Learning with Contrastive Predictive Coding". From this perspective, what the authors are proposing here is not really CPC, but a combination of CPC and APC. In a way, you're doing yourself an injustice by just calling this "CPC", since you've invented a new way of doing representation learning that seems to have the best of both worlds!
__________
Here's another thing I wonder about. The implicit distribution used in APC is a Laplace distribution because we're using an L1 loss. We could also use an L2 loss, in which case it would be a Gaussian distribution. Does that make a difference? Have you tried it? I'd love to see that, if it's not too expensive for you to run the experiment. (It should just be a one-line change: `(output - target)**2' instead of `| output - target |'.)

[1] - the CPC paper: https://arxiv.org/pdf/1807.03748.pdf
[2] - the APC paper: https://arxiv.org/pdf/1904.03240.pdf

---

### Decision · Program_Chairs · 2020-07-01

**Decision:**

Accept

**Comment:**

Dear author(s),

Thank you very much for your submission at the ICML2020@SaS workshop (https://icml-sas.gitlab.io/). Based on the scores assigned by the reviewers, we are happy to notify you that your paper was accepted for the workshop.

Please, address the comments of the reviewers and submit the camera-ready version by July 8. We ask the authors to record a 15min video for your talk. At the workshop, we will have the pre-recorded video as well as a live QA session. It is important to keep this time limit, otherwise, your talk will be automatically cut. The deadline for uploading the video is July 8. The detailed instructions for uploading will follow.

Feel free to contact us for any questions!

Best,

The ICML20@SaS organizers:
Mirco Ravanelli
Titouan Parcollet
Dmitriy Serdyuk
Devon Hjelm
Bhuvana Ramabhadran